# Strength of Sense of Coherence among Nurses and the Relationship between Socio-Demographic and Work-Related Factors

**DOI:** 10.3390/ijerph20105786

**Published:** 2023-05-11

**Authors:** Martina Smrekar, Lijana Zaletel-Kragelj, Sanja Ledinski Fičko, Snježana Čukljek, Biljana Kurtović, Ana Marija Hošnjak, Alenka Franko

**Affiliations:** 1Department of Nursing, University of Applied Health Sciences Zagreb, Mlinarska Cesta 38, 10000 Zagreb, Croatia; sanja.ledinski-ficko@zvu.hr (S.L.F.); snjezana.cukljek@zvu.hr (S.Č.); biljana.kurtovic@zvu.hr (B.K.); anamarija.hosnjak@zvu.hr (A.M.H.); 2Chair of Public Health, Faculty of Medicine, University of Ljubljana, Zaloska 4, 1000 Ljubljana, Slovenia; lijana.zaletel-kragelj@mf.uni-lj.si (L.Z.-K.); alenka.franko@siol.net (A.F.); 3Clinical Institute of Occupational Medicine, University Medical Centre Ljubljana, Grabloviceva 42, 1000 Ljubljana, Slovenia

**Keywords:** sense of coherence, nurses, work, stress

## Abstract

Sense of coherence (SOC) occupies the central place within the salutogenic model. It is an important contributor to the development and maintenance of people’s health. This study aimed to assess the strength of sense of coherence (SOC) among nurses and the relationship between the strength of SOC and socio-demographic and work-related factors. A cross-sectional study was conducted in 2018. Linear regression was used to describe strength of association between SOC and socio-demographic and work-related factors. A total of 713/1300 nurses completed an SOC-29-item questionnaire for the assessment of SOC. The mean value for total SOC score (SOCS) was 145.0 points (SD 22.1, range 81–200). The results of the multivariate linear regression revealed statistically significant positive associations between SOCS and age (>40 years), level of education (master of nursing and bachelor of nursing), and transportation mode by car. Our study suggested SOC as an important and influential health-promoting personal resource of nurses which might offer protection regarding work-related stress.

## 1. Introduction

The theory of salutogenesis was introduced in 1970 by the American Israeli medical sociologist Aaron Antonovsky [1]. In contrast to the dichotomous classification of individuals into categories of sick or well, Antonovsky has seen health as a healthy/disease continuum and sense of coherence (SOC) as a central resource responsible to determine an individual’s movement on that continuum. According to Antonovsky, individuals are often exposed to changes in everyday life functioning, which can result in the occurrence of stress and consequently the development of the disease [1,2]. Thus, SOC has been seen as a resource that enables individuals to manage tension by identifying and mobilizing resistance resources for effective coping in stressful situations [3]. These resources are named as generalized and specific resistance resources (GRRs/SRRs). GRRs can be described as biological, material, and psychosocial factors found within the individuals and their environment such as material resources, knowledge and intelligence, ego identity, coping strategies, social support, and cultural stability, while SRRs are resources relevant to particular circumstances [2].

Various studies have shown that nurses all over the world are exposed to stressful situations in nursing on a daily basis [3,4,5,6,7]. Eriksson et al. (2019) stated that nurses have difficulties in managing stress and that it is important to address problems of stress in nursing [3]. It was suggested that occupational stress affects nurses’ health-related quality of life negatively and has an influence on patient outcomes [6]. Wright et al. (2014) stated that nurses are more likely than other healthcare professionals to experience ill health due to daily basis exposure to various stressful situations [7]. According to Eriksson et al. (2019), research findings on SOC in the nursing profession are limited [3]. Evidence has shown that individual and work-related factors have an influence on nurses’ SOC and health. Some of these factors are age, gender, marital status, level of education, years of work experience, type of employment, shift work, work department, job satisfaction, and work engagement [8]. Eriksson et al. (2019) revealed that age was a significant predictor of SOC score among nurses [3]. Moreover, research findings from various studies demonstrated a positive association between marital status [9], education [10], work experience [9], job satisfaction [11], and SOC among nurses. Several studies have reported that strong SOC is an important factor that helps nurses in coping with stress [3,12,13]. A strong SOC is a protective factor for nurses’ depressive state, burnout, and job dissatisfaction [8] and it is positively associated with nurses’ good mental and physical health status [8,9,14]. Research findings have shown that SOC is also as an important and suitable intervention-targeted outcome to consider in managing workability difficulties for nurses in clinical practice [15]. Additionally, a systematic review on SOC among nurses pointed out the need for further research on the impact of socio-demographic characteristics on SOC [8]. The SOC could be considered as an important factor of adjustment to stress and workload in nursing profession. The findings of this research study could serve as evidence for evidence-based planned public health measures for maintaining/enhancing good health among nurses. The aim of the present study was to assess the strength of SOC among nurses and the relationship between the strength of SOC and socio-demographic and work-related factors.

## 2. Materials and Methods

### 2.1. Study Population and Design

The research was meant to be carried out on the entire population of the selected hospital and not on a sample. A total population of 1465 nurses of different profiles (registered nurses, bachelor of nursing, master of science in nursing) employed in different departments of the University Hospital Centre (UHC) in Croatia were considered for the inclusion in the study. Consequently, it was not necessary to choose a method to select a sample and define its size. The only exclusion criterion was absence from workplace at the time of the survey. Due to various absences such as sick leave, annual leave, and study leave, 165 nurses were excluded and consequently, 1300 were invited to participate in the study.

### 2.2. Study Procedure

After obtaining the approval of the Ethics Committee of UHC, the principal investigator organized a meeting with the head nurses of each clinic at the UHC and presented them with the aim and objectives of the research and the research protocol. Afterwards, the principal investigator distributed the study questionnaires to nurses in all hospital departments. Special identification codes unique to each participant were used in order to assure anonymity.

### 2.3. Ethical Coniderations

An application for ethical review of the proposed research project was submitted to the UHC ethics committee. All documentation required was submitted by the applicant (signed and dated application form, the protocol of the proposed research, copies of the instruments, copy of the information for respondents, and the informed consent form). Afterwards, the applicants were invited to present the proposal. After considering the application, the study protocols were approved by the UHC (code: EP–7811/16-19). The study was carried out in accordance with the ethical principles of the Helsinki Declaration. Nurses were informed that participation in the study was voluntary and anonymous. Informed consent was obtained from each nurse before enrolment.

### 2.4. Description of the Study Instrument

The Orientation to Life Questionnaire (SOC-29 scale) was used for the measurement of the SOC. The scale consists of 29 items with a seven-point response scale. SOC score (SOCS) is a summary measure obtained by summing up the scores of individual responsesto all items of SOC-29, ranging from 29 to 203 points. A higher SOCS suggested a stronger SOC [2]. In this study, we used the Croatian version of the SOC scale (SOC-29 CRO), which proved to be a reliable and valid instrument for being used on the population of Croatian nurses [16]. In the analysis, SOCS was the observed outcome.

The data were collected using socio-demographic and work-related factors on the questionnaire. The socio-demographic questionnaire items were age (nurses were asked to write their age in years), gender (male, female), marriage status (married, divorced, widower, single, life partnership), level of education (secondary school education, bachelor of nursing, master of nursing), ongoing education (no, yes). The work-related questionnaire items were work length (nurses were asked to write their years of work experience in years), department (nurses were asked to write the name of department where they work), transportation mode to work (nurses were asked to select one or more of the four choices: public transport, personal car, train/bus, on foot), and time to come to work (0–15 min, 15–30 min, 30–60 min, 60–90 min, 90 or more minutes).

During the statistical analysis, age was recorded in two categories: ≤40 and >40 years; gender in two categories: female and male; marital status in two categories: married and other; educational level in three categories: secondary school education, bachelor of nursing and master of nursing; ongoing education in two categories: yes and no; work length in three categories: <1 year, 1–20 years, and >20 years; work department in two categories: departments with special demands (polyclinic unit, oncology and hematology unit, psychiatry unit, pediatrics unit) and the other; time to come to work in two categories: ≤30 min and >30 min; transportation mode in three categories: transportation only by car, on foot only, and other.

### 2.5. Statistical Analysis

In the statistical analysis, standard descriptive statistics were first performed and reported as frequencies, percentages, means, standard deviation, and minimum and maximum value. The SOCS was used in the analysis. The association between SOC and socio-demographic and work-related factors was assessed univariately and multivariately by using the linear regression method; *p*-values of 0.05 or less were considered as significant. The statistical analysis was performed with the SPSS software, version 21.0 (SPSS Inc., Chicago, IL, USA).

## 3. Results

### 3.1. Description of the Study Group

A total of 713/1300 nurses participated in the study (response rate: 54.7%). The mean age of participants was 38.4 years (SD 12.5, range 19–65), and the mean work length was 17.48 years (SD 12.8, range 0–45). The mean value of total SOCS was 145.0 points (SD 22.1, range 81–200). The mean values for SOCS regarding socio-demographic and work-related factors are presented in Table 1.

Regarding socio-demographic and work-related factors, the differences in SOCS between different categories were observed. The mean value for SOCS was statistically significant for respondents aged >40 years compared to respondents aged ≤40 years (F = 19.721; *p* < 0.001); married participants compared to others not married (F = 9.607; *p* = 0.002); participants with a bachelor of nursing and master of nursing level of education compared to secondary school education (F = 5.047; *p* = 0.007); participants with >20 years of work experience compared to participants with <1 year of work experience (F = 7.876; *p* < 0.001); participants working in departments with special demands (polyclinic unit, oncology and hematology unit, psychiatry unit, pediatrics unit) compared to other departments (F = 5.567; *p* = 0.019); participants who spend ≤30 min to arrive at work compared to participants who spend >30 min to arrive at work (F = 3.950; *p* = 0.047), and participants who arrive to work by car compared toother modes (F = 6.585; *p* < 0.001). The mean value for SOCS was not statistically significant for men compared to women (F = 0.447; *p* = 0.504); participants who were not involved in ongoing education compared to those who were involved in ongoing education (F = 0.284; *p* = 0.594) (Table 1).

### 3.2. Results of Univariate Analysis

The results of the univariate linear regression revealed statistically significant positive associations between SOCS and age (>40 years compared to ≤40 years) (b = 7.449, 95% CI 4.156–10.742, *p* < 0.001), meaning that if age increased by one point, the SOCS improved for 7.449 points. Statistically significant positive associations were also observed between SOCS and marital status compared to others not married (b = 5.183, 95% CI 1.900–8.466, *p* = 0.002), level of education (master of nursing compared to secondary school) (b = 12.364, 95% CI 3.803–20.926, *p* = 0.005), work length (>20 years compared to <1 year) (b = 9.747, 95% CI 0.250–19.244, *p* = 0.044), work departments with special demands compared to other departments (b = 4.913, 95% CI 0.824–9.002, *p* = 0.019), time to come to work (≤30 min compared to >30 min) (b = 3.527, 95% CI 0.043–7.011, *p* = 0.047), and transportation mode by car compared to other modes (b = 2.929, 95% CI 2.929–9.836, *p* < 0.001) (Table 2). On the other hand, in the univariate analysis, female gender compared to male, level of education (bachelor of nursing compared to secondary school, attending ongoing education compared to not attending ongoing education), work length (1–20 years compared to <1 year), and transportation mode on foot only compared to other modes did not demonstrate to be statistically significant (Table 2). Univariately, age (>40 years) was the strongest predictor of SOCS solely explaining 2.8% of the variance of the SOCS (R^2^ = 0.028, *p* < 0.001), followed by work length (>20 years) explaining 2.2% (R^2^ = 0.022, *p* = 0.044) of the variance in SOCS; transportation mode by car explaining 1.9% of the variance in SOCS (R^2^ = 0.019, *p* < 0.001); marital status explaining 1.4% (R^2^ = 0.014, *p* = 0.002) of the variance in SOCS; master of nursing level of education explaining 1.4% (R^2^ = 0.014, *p* = 0.005) of the variance in SOCS; high demand work departments explaining 0.8% (R^2^ = 0.008, *p* = 0.019) of the variance in SOCS, and time to come to work explaining 0.6% (R^2^ = 0.006, *p* = 0.047) of the variance in SOCS (Table 2). The details are presented in Table 2.

### 3.3. Results of Multivariate Analysis

Complete data for all variables entering the multivariate model were available for 657 participants. The results of the multivariate linear regression revealed statistically significant positive associations between SOCS and age (>40 years compared to ≤40 years) (b = 9.219, 95% CI 2.028–16.410, *p* = 0.012), level of education (master of nursing compared to secondary school) (b = 10.515, 95% CI 1.917–19.114, *p* = 0.017), transportation mode by car compared to other modes (b = 5.941, 95% CI 2.466–9.417, *p* = 0.001) (Table 3). Additionally, although the bachelor of nursing level of education tested univariately and was not statistically significantly different from secondary school education, it appeared to be statistically significantly different in the multivariate model (b = 4.393, 95% CI 0.889–7.898, *p* = 0.014) (Table 3).

In comparison to the results of the univariate analysis, in the multivariate analysis, age (>40 years) showed an increased strength of association with SOCS (b = 7.449, *p* < 0.001 versus b = 9.219, *p* = 0.012), meaning that if age increased by one point, the SOCS improved for 9.219 point. On the contrary, in the multivariate analysis, education (master of nursing) (b = 12.364, *p* = 0.005 versus b = 10.515, *p* = 0.017) and transportation mode by car (b = 6.382, *p* < 0.001 versus b = 5.941, *p* < 0.001), showed a decreased strength of association with SOCS compared to the univariate analysis (Table 2 and Table 3). 

The percentage of variability of SOCS that could be explained increased in the multivariate model to 7.9% (R^2^ = 0.079, *p* = 0.012). However, in the multivariate regression analysis, female gender compared to male, marital status (married compared to not married), attending ongoing education compared to not attending ongoing education, work length (1–20 years compared to <1 year and >20 years compared to <1 year), work departments with special demands compared to others, time to come to work (≤30 min compared to >30), and transportation mode on foot only compared to other modes did not demonstrate to be statistically significant (Table 3). The details are presented in Table 3.

## 4. Discussion

The results of our research showed statistically significant positive associations between SOC (assessed by SOCS) and age, level of education, and transportation mode to the workplace, but not between gender, work length, work department, and time to come to work. The mean value for SOCS in our research was similar to the findings of the study conducted among nurses employed in a Portugal hospital [17]. A higher SOCS was reported among hospital nursing managers [10], while among nurses employed in the Poland hospital, it was considerably lower [14]. It has been suggested that a higher SOCS protects people from stress [18] due to the fact that they become more resilient under stress [19] and have the capacity to respond to stressful situations more effectively [20]. In addition, it was suggested that workplace stressors in the nursing profession could be managed by a strong SOC [3,12]. A strong SOC can be seen as a coping resource that individuals possess and which enables them to use different strategies for problem solving and managing stressful life events [21]. Therefore, a strong SOC is recognized as an important factor in the maintenance of health [22]. According to Malagon-Aguilera et al. (2019), a strong SOC could help nurses to better manage occupational stress, which could result in better health and greater work engagement [12]. On the contrary, it has been suggested that a weak SOC could be consider as a vulnerability factor for nurses [8]. It has also been suggested that a weak SOC could be a predictor of poor workability and improving SOC could significantly increase the workability among nurses [15].

The present research revealed that participants aged >40 years have a higher SOCS on average. This finding is consistent with the result of a previous study [3]. Furthermore, in the present research, age (>40 years) was demonstrated to be a significant predictor of SOCS in the univariate analysis and its power even increased in a multivariate model. According to Eriksson et al. (2019), older age helps hospital nurses to identify strategies to manage work-related stress [3]. Similar to our results, a significantly stronger SOC was reported among older adults in several studies in other population groups [23,24,25]. The literature states that the level of SOC increases more over time for younger adults than for older adults [23]. In addition, it has been suggested that the SOC scores increase with increasing age [26]. Sardu et al. (2012) demonstrated that age (≥30 age) was significantly associated with a strong SOC among the Italian general population [27]. Age and SOC were positively correlated in a study among adult patients with Type 2 diabetes where it was suggested that the SOC increases with age and that age can be considered as an important factor to improve the SOC [28]. Contrary to our findings, previous studies have shown that there is no significant relationship between age and SOC [10,29,30]. 

The current research demonstrated that married participants have a higher SOCS on average. This is consistent with the results from the study conducted among hospital nurses in Japan, which showed a positive association between marital status and SOC [9]. Previous research has suggested that married individuals often report a high level of well-being [31] and life satisfaction [32]. Furthermore, married individuals are reported to share a common income, which contributes to the financial security of an individual [33]. Thus, marriage may act as a potential GRR for strengthening an individual’s SOC [34]. Contrary to the findings of our study, Kretowicz and Bieniaszewski (2015) did not find a significant relationship between SOC and marriage status [10].

The results of the present research revealed that participants with a bachelor’s and master’s degree in nursing have a higher SOCS on average. A similar finding was reported in the study by Kretowicz and Bieniaszewski (2015), who suggested that obtaining a master’s degree in nursing substantially raises the global SOC [10]. According to Masanotti et al. (2020), the progress in academic education could elevate the global SOC score [8]. On the contrary, in the study by Eriksson et al. (2019), the level of education did not demonstrate a significant association with the SOC [3]. 

In the current research, the results of the univariate analysis showed that participants with >20 years of work experience have a higher SOCS on average. A similar result was reported in the study by Miyata et al. (2015) [9]. On the contrary, Erikson et al. (2019) did not find significant associations between length of work experience and SOC [3]. According to Eriksson et al. (2019), longer work experience provides hospital nurses with a sense of security in their professionalism [3]. The categorization of the variable “work length” was based on the general observations of nurses’ professional development. In their first year of employment, nurses intensively adapt to the new situation in their professional environment. In this period, SOC is related to how it was strengthened in previous periods of life, and the possibilities for additional SOC strengthening are therefore minimal. During the next 20 years, nurses, while gaining work experience, more or less develop defense mechanisms, including a stronger or weaker SOC, while in the next period, after about 20 years, these mechanisms are already developed and already sufficiently well formed.

The findings of the present research revealed statistically significant associations between work department and SOCS. More specifically, nurses working in the hospital departments with special demands (polyclinic unit, oncology and hematology unit, psychiatry unit, pediatrics unit) had a higher SOCS on average compared to nurses working in the internal unit, surgery unit, operating room, intensive care unit, gynecology unit, dermatology unit, emergency unit, and ophthalmology unit. Departments with special demands are different from other departments due to a specific work organization. Nurses who work in these departments are confronted with many challenges and unexpected situations. In such an environment, they often need to react quickly, and gain strength and experience faster than in other hospital departments. According to Erikson et al. (2019), the nurses’ working environment is unpredictable [3]; therefore, it is important for nurses to have a strong SOC in order to effectively manage various stressful situations [12]. In the study by Eriksson et al. (2019), the work department did not demonstrate significant associations with SOC [3]. 

A statistically significant association between ≤30 min time spent traveling to work and SOCS was revealed in the univariate analysis. Participants who spent ≤30 min on traveling to work had a higher SOCS on average. This could be explained by the fact that hospital nurses who spend less time on traveling to work have more free time which they can dedicate to family or other activities. Our finding is in agreement with the results of a study by Turchi et al. (2019), which showed that a longer travel time to work had a negative influence on hospital nurses’ psychological well-being [35]. It has been observed that by increasing travel time, the perceived stress of commuting increases [36]. 

The results of the present research revealed a statistically significant association between transportation to work by a car and SOCS, which was shown in both univariate and multivariate analyses. More specifically, participants who used a car as a transportation mode to the workplace had a higher SOCS on average. The explanation could be that public transportation can be stressful [37]. In addition, owning a car may indicate a better financial status. 

### 4.1. Study Strengths

UHC employs a diverse profile of healthcare professionals and a large number of nurses, which allowed us to investigate work-related issues that are important for the nursing profession and nurses’ well-being. Thus, the results of this study could be applied to hospitals around the world. Second, the present study provided important information and knowledge about the relationship between SOC and a wide range of different socio-demographic and work-related factors contrary to the previous studies which investigated the relationship between SOC and only a limited number of socio-demographic and work-related factors. 

### 4.2. Study Limitations

The present study has some potential limitations. First, given the limitations of the study’s design, any conclusions about the cause-and-effect association between study variables and SOCS were unable to be reached. Nevertheless, the present study gives valuable information which is needed to plan further research in this area.

Second, one could argue about the departments with special demands (polyclinic unit, oncology and hematology unit, psychiatry unit, pediatrics unit). The working environment for nurses in these departments is unpredictable and requires a specific work organization. Although, nurses working in these departments are confronted with many unexpected stressful situations, workload and stress pattern may different. 

Third, one could argue about the fact that the present research was conducted in 2018, and a secondary analysis was not conducted. However, the results of the present research still contribute valuable information in the area of research related to the SOC and salutogenic theory. Concerning nurses, the SOC could be helpful in understanding the necessary support for nurses.

### 4.3. Implications for Public Health

The results of the present study could contribute to new insights in the area of research related to SOC and salutogenic theory. Concerning nurses, SOC could be helpful in understanding the necessary support for nurses. Implementation of programs to increase SOC among nurses is required in order to preserve and maintain the health of nurses. It would be crucial to integrate the salutogenic approach in all public health policy making.

### 4.4. Suggestions for Future Research

Studies estimating the effect of interventions for enhancing the SOC among nurses are needed. Furthermore, more studies are needed to establish whether increasing SOC among nurses is associated with patient outcomes.

## 5. Conclusions

Our study suggests SOC as an important and influential health-promoting personal resource for nurses, which might offer protection regarding work-related stress. Given the significance of maintaining health and positive work engagement among nurses, SOC could be seen as an effective mechanism for coping with stressful situations. The implementation of interventions to strengthen nurses’ SOC is important and crucial for the nursing profession. 

## Figures and Tables

**Table 1 ijerph-20-05786-t001:** Mean values for SOCS regarding socio-demographic and work-related factors (687–692).

Factor	Category	n	n_cat_	Mean	SD	Min	Max
**SOCIO-DEMOGRAPHIC FACTORS**							
Gender	Female	692	614	144.6	21.8	81	200
	Male		78	146.4	24.4	89	194
Age (years)	≤40	691	397	141.7	20.8	87	194
	>40		294	149.1	23.2	81	200
Marital status	Married	691	365	147.3	22.5	81	200
	Other		326	142.1	21.3	89	193
Level of education	Secondary school	692	415	143.2	22.5	81	200
	Bachelor of nursing		250	146.4	20.1	87	192
	Master of nursing		27	155.5	26.3	101	199
Ongoing education	No	691	547	145.0	22.3	81	199
	Yes		144	143.9	21.2	95	200
**WORK-RELATED FACTORS**							
Work length (years)	<1	692	22	138.9	21.1	96	176
	1–20		382	142.3	21.1	87	194
	>20		288	148.7	22.9	81	200
Work department	Departments with special demands	687	141	148.7	23.8	81	199
	Other		546	143.8	21.6	86	200
Time to come to work (minutes)	≤30	689	234	147.1	21.6	81	199
	>30		455	143.6	22.3	86	200
Transportation mode	By car only	686	245	148.7	22.5	89	200
	On foot only		19	144.8	15.0	113	174
	Other		422	142.3	21.8	81	200

Key: SOCS—sense of coherence score; SD—standard deviation.

**Table 2 ijerph-20-05786-t002:** Results of univariate linear regression analysis of association between SOCS and socio-demographic and work-related factors (687–692).

	Category				95% CI for b			
Factor	Observed	Reference	n	b	Lower	Upper	*p*	R^2^
**SOCIO-DEMOGRAPHIC FACTORS**								
Gender	Female	Male	692	−1.775	−6.990	3.439	0.504	0.001
Age (years)	>40	≤40	691	7.449	4.156	10.742	<0.001	0.028
Marital status	Married	Other	691	5.183	1.900	8.466	0.002	0.014
Level of education	Bachelor of Nursing	Secondary school	692	3.246	−0.205	6.697	0.065	0.014
	Master of Nursing	Secondary school		12.364	3.803	20.926	0.005	0.014
Ongoing education	Yes	No	691	−1.101	−5.156	2.954	0.594	0.000
**WORK-RELATED FACTORS**								
Work length (years)	1–20	<1	692	3.340	−6.074	12.753	0.486	0.022
	>20	<1		9.747	0.250	19.244	0.044	0.022
Work department	Departments with special demands	Other	687	4.913	0.824	9.002	0.019	0.008
Time to come to work (minutes)	≤30	>30	689	3.527	0.043	7.011	0.047	0.006
Transportation mode	By car only	Other	692	6.382	2.929	9.836	<0.001	0.019
	On foot only	Other		–0.539	−7.544	12.622	0.621	0.019

Key: SOCS—sense of coherence score; b—unstandardized coefficients; CI—confidence interval; R^2^—R-square, coefficient of determination.

**Table 3 ijerph-20-05786-t003:** Results of multivariate analysis of association between SOCS and socio-demographic and work-related factors (n = 657).

	Category			95% CI for b		
Factor	Observed	Reference	b	Lower	Upper	*p*
Gender	Female	Male	−3.207	−8.524	2.110	0.237
Age (years)	>40	≤40	9.219	2.028	16.410	0.012
Marital status	Married	Other	3.370	−0.121	6.860	0.058
Level of education	Bachelor of Nursing	Secondary school	4.393	0.889	7.898	0.014
	Master of Nursing	Secondary school	10.515	1.917	19.114	0.017
Ongoing education	Yes	No	2.705	−1.616	7.026	0.219
Work length (years)	1–20	<1	1.198	−9.417	11.814	0.825
Work length (years)	>20	<1	−0.514	−13.194	12.166	0.937
Work department	Departments with special demands	Other	2.918	−1.144	6.979	0.159
Time to come to work (minutes)	≤30	>30	2.401	−1.164	5.966	0.186
Transportation mode	By car only	Other	5.941	2.466	9.417	<0.001
	On foot only	Other	1.594	−8.540	11.728	0.758

Key: b—unstandardized coefficients; CI—confidence interval.

## Data Availability

The data presented in this study are available on request from the corresponding author.

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
