# Peer review of "Strength of Sense of Coherence among Nurses and the Relationship between Socio-Demographic and Work-Related Factors"

_ijerph, 2023, doi:10.3390/ijerph20105786_

Round 1

Reviewer 1 Report

I would like to congratulate the team of authors for the study carried out on the Sense of Coherence (SOC) as an important factor in the promotion of people's health and as a protective resource for nurses in the face of work-related stress.

The study is well designed and the methods and tools used are described, although being a cross-sectional study it has the limitations of such studies.

In the attached document, I would like to make my recommendations.

Best regards.

Author Response

We thank Reviewer 1 for his comments on our manuscript!

Thank you!

Reviewer 2 Report

Thank you for the opportunity to review this paper. This study investigated the strength of sense of coherence (SOC) among nurses and the relationship between the SOC and socio - demographic and work - related factors. The overall design, results, tables and discussion are well described. However, some concern should be clearly discussed.
1. Table 1, work lengths were defined to three groups which were very different in range (less than 1, 1 to 20 and more than 20 years). The investigators should discuss why choosing this range.
2. Table 1, transportation mode in the category “other” were larger than by car and on foot. The investigators should show some essential modes or top three transportation modes.
3. The investigators collected the name of the departments where the subjects work but presented as department with special demands (polyclinic unit, oncology and hematology unit, psychiatry unit, pediatrics unit). Workload and stress pattern maybe different (e.g., psychiatric nurse may face with more difficult situation or more complicated issue than pediatric nurse) and may interfere with the analysis hence this should be mentioned in the discussion.
4.
Given the limitations of the study's design, it was unable to make any conclusions about the cause-and-effect association between study variables and SOCS. This needs to be addressed as the study's primary limitation.  

Author Response

We thank Reviewer 2 for his comments on our manuscript!

Thank you!

Reviewer 3 Report

Dear authors,

Thank you for submitting this article and congratulations on your work. Please find below comments/suggestions for improving the article.

- In the introduction, the contribution of the research to clinical practice should be further defined.
- The study was conducted in 2018. Please identify if a secondary analysis was performed and elaborate why the results are still relevant. Please also indicate the limitations of the study.
- Describe in detail on what basis the sample size was determined.
- Add the ethical aspect of the research.
- Identify which statistical tests were used.
- Define what the SOCS values mean.

Author Response

We thank Reviewer for his comments on our manuscript!

Thank you!
